# The Brazilian COVID-19 vaccination campaign: a modelling analysis of sociodemographic factors on uptake

Sabrina L Li [1,2] Carlos A Prete Jr [3] Alexander E Zarebski [4,5] Andreza Aruska de Souza Santos [6,7] Ester C Sabino [8] Vitor H Nascimento [3] Chieh-Hsi Wu [9] Jane P Messina [2,10]

SLL, CAPJ and AEZ contributed equally.

**Correspondence to**
Dr Sabrina L Li;
sabrina.li@nottingham.ac.uk

## ABSTRACT

**Objective** Dose shortages delayed access to COVID-19 vaccination. We aim to characterise inequality in two-dose vaccination by sociodemographic group across Brazil.

**Design** This is a cross-sectional study.

**Setting** We used data retrieved from the Brazilian Ministry of Health databases published between 17 January 2021 and 6 September 2021.

**Methods** We assessed geographical inequalities in full vaccination coverage and dose by age, sex, race and socioeconomic status. We developed a Campaign Optimality Index to characterise inequality in vaccination access due to premature vaccination towards younger populations before older and vulnerable populations were fully vaccinated. Generalised linear regression was used to investigate the risk of death and hospitalisation by age group, socioeconomic status and vaccination coverage.

**Results** Vaccination coverage is higher in the wealthier South and Southeast. Men, people of colour and low-income groups were more likely to be only partially vaccinated due to missing or delaying a second dose. Vaccination started prematurely for age groups under 50 years which may have hindered uptake in older age groups. Vaccination coverage was associated with a lower risk of death, especially in older age groups (ORs 9.7 to 29.0, 95% CI 9.4 to 29.9). Risk of hospitalisation was greater in areas with higher vaccination rates due to higher access to care and reporting.

**Conclusions** Vaccination inequality persists between states, age and demographic groups despite increasing uptake. The association between hospitalisation rates and vaccination is attributed to preferential delivery to areas of greater transmission and access to healthcare.

## INTRODUCTION

During the COVID-19 pandemic, Brazil experienced the third highest number of confirmed COVID-19 deaths globally. Despite having a universal healthcare system that is experienced with managing public health emergencies,[1] Brazil was unable to contain COVID-19[2] and intensive care unit beds consistently exceeded capacity in many states.[3] This was complicated by the emergence of the Gamma (P.1) and Omicron

## STRENGTHS AND LIMITATIONS OF THIS STUDY

⇒ A key strength of this study is that data were collected from the Brazilian Ministry of Health's national immunisation system which included information on every individual that participated in the vaccination programme.

⇒ We combined spatial data on vaccination, cases of severe acute respiratory illnesses and confirmed COVID-19 cases (due to low rates of COVID-19 testing in Brazil), socioeconomic conditions and recent population estimates from 2020 to account for excess COVID-19 deaths.

⇒ We built a spatially explicit model to robustly infer vaccination uptake and health risk across Brazil.

⇒ We developed a vaccination Campaign Optimality Index, which describes how vaccination was rolled out, estimating the extent to which older groups were and were not prioritised during the campaign.

⇒ Our results are limited by being unable to retrieve the vaccination status of individuals infected with COIVD-19, and therefore, relied on overall vaccination for each age group and municipality.

(B.1.1.529) variants.[4] The COVID-19 vaccination campaign was launched across Brazil on 17 January 2021 and initially prioritised healthcare workers, individuals aged 90 years or older, indigenous communities and institutionalised individuals. Four types of vaccines have been offered during the pandemic. However, only Coronavac (Sinovac, China) and Covishield AZD1222 (Oxford-AstraZeneca, UK) vaccines were offered during the early phase of the campaign. Vaccine shortages and delays have caused unanticipated interruptions in the administration of doses after the first dose.[5]

Equitable and ethical administration of vaccines should be strived for; however, this remains an ongoing challenge.[6 7] During the campaign, dose allocation was determined by the total population of each state without adjusting for demographic

differences. While older age is associated with varying risk of SARS-CoV-2 infection,[8] it is associated with increasing risk of hospitalisation and death.[9 10] Thus, understanding vaccine uptake across age groups is critical for reducing mortality and health inequalities, and optimising the distribution of limited vaccine doses. More importantly, little is known about how the vaccination programme impacts age-specific hospitalisations and COVID-19-related death rates.[11] To optimise vaccine distribution, further investigation is needed to understand the age-specific reduction in the risk of severe COVID-19 and death, especially among those most vulnerable.

We investigated inequalities in vaccination coverage across Brazil in 2021 and assessed how factors such as age, sex, race and socioeconomic status, influenced uptake. We found several states delayed full vaccination for older populations to prioritise allocation for younger individuals. Among the population that was vaccinated, we estimated the proportion of individuals that did not take the second dose, finding large variation with income, race, sex and across states. We also estimated the effect of vaccination on the risk of hospitalisation and death from COVID-19 and its associations with socioeconomic conditions. We further assessed the effects of vaccination coverage on death across age groups.

## METHODS
### Patient and public involvement
None.

### Data sources
#### Vaccination information
Patient-level vaccination data notified between 17 January 2021 and 6 September 2021 were retrieved from the Brazilian Ministry of Health's national immunisation system, Sistema de Informação do Programa Nacional de Imunizações (SI-PNI) (https://opendatasus.saude.gov.br/dataset/covid-19-vacinacao). For every individual that received a dose, we retrieved their age, sex, municipality and state of residence, vaccination facility, dose number (eg, first or second) and the vaccine type. We did not consider the booster dose campaign in our study, which started on 15 September 2021.

#### Cases of severe acute respiratory illness
We retrieved data on hospitalisations and deaths in patients with severe acute respiratory illness (SARI) between 1 March 2020 and 16 September 2021, from the SIVEP-Gripe database, curated by the Ministry of Health of Brazil. Patient information on age, sex and municipality of residence was included. All SARI cases and deaths were notified in the SRAG database regardless of hospitalisation. SARI can be caused by SARS-CoV-2 and is defined by the Ministry of Health as influenza-like illness plus one of the following: dyspnoea, persistent chest pain or hypoxia. We selected SARI hospitalisations and

deaths, disregarding non-hospitalised individuals that did not die. All SARI cases confirmed to be caused by other respiratory viruses were excluded from the analysis. We included both confirmed COVID-19 cases and SARI cases with unknown aetiology, as those were likely related to COVID-19 but not lab confirmed.[2 12] This might be due to two reasons, such as: the low rates of COVID-19 testing in Brazil, especially in poorer areas as shown by de Souza *et al*[12] ; and high proportion of false negative tests that were likely applied after the time window where SARS-CoV-2 is detectable by PCR. From all SARI cases with unknown or missing aetiology with symptom onset during the study period, 24.9% did not have a PCR test result reported in SIVEP-Gripe and most of the remaining 75.1% had likely a false negative test result.

#### Socioeconomic conditions and population size
Measurements of socioeconomic indicators at the municipality level were retrieved from the latest population census (2010) compiled by the Brazilian Institute of Geography and Statistics.[13] We selected indicators based on their relevance to the social determinants of health, such as household income per capita, proportion of residents with only a primary education or lower and unemployment rates. Projected distributions of age and sex for the population in 2020 were obtained for each municipality and state.[14] Incomes were converted from Reais 2010 to US$ purchase power parity (PPP) 2010 using the conversion rate of 1.388, available at https://data.oecd.org/conversion/purchasing-power-parities-ppp.htm.

### Data analysis
#### Age group-specific vaccination coverage
We estimated the proportion of the population vaccinated with at least one dose using the SI-PNI data and the projected population in each municipality, and for each age group within that municipality: under 20, 20–29, 70–79 and 80 years and above.[14]

To compare the levels of first dose coverage between older and younger populations, we defined a Campaign Optimality Index (COI). The COI for age group A at threshold T is denoted as $\eta\left(A, T\right)$. The COI measures the proportion of the subsequent (older) age group at the time when the age group $A$ first reached its threshold relative coverage of $T$. The relative coverage at time $t$ is defined as the coverage at time $t$ divided by the final coverage on 6 September 2021. To compute the optimality index for threshold $T$ and age group $A = \left[a_{\min}, a_{\max}\right]$ (which includes individuals aged between $a_{\min}$ and $a_{\max}$), we calculated the date when the relative coverage in that age group reached the threshold $T$ and then defined $\eta\left(A, T\right)$ as the relative coverage of subsequent age group $A_{\text{older}} = \left[a_{\min} + 10, a_{\max} + 10\right]$. We use relative coverages instead of the absolute coverages to avoid biasing COI due to errors in population estimate.

If vaccination was only offered to an age group $A$ when everyone in the subsequent age group was vaccinated, then the COI would be 1. Smaller values of the COI for

a particular age group indicate that fewer people in the older age group were vaccinated. In an ideal scenario, the vaccination campaign for younger groups begins only after older age groups have achieved high coverage. In this case, the coverage of the older group $A_{older}$ should be close to 100% when the coverage of the younger group reaches $T$, in which case $\eta(A, T) = 1$. The COI was computed for all age groups and using thresholds of 25%, 50% and 75%.

## Population that missed their second dose

We estimated the proportion of the population that delayed the second dose, that is, the proportion did not receive their second dose within the suggested time frame (21–28 days interval for Coronavac, and 90 days for all other approved two-dose COVID-19 vaccines). To estimate the proportion of the population that missed the second dose entirely, we determined (1) the proportion of all first dose vaccinated individuals that did not have a second dose recorded, and (2) from this, selected only those for which the time interval between the first dose and the date of the last entry in the dataset (6 September 2021) is higher than the suggested time interval between doses for the corresponding vaccine plus a tolerance of

30 days. In other words, this is the proportion of individuals that did not take the second dose and whose second dose is delayed by at least 30 days. Both proportions were estimated for each age, sex, race and municipality. We excluded the single-dose vaccine of Janssen-Cilag from this analysis.

## Risk of COVID-19 death by age group

We used logistic regression to estimate the probability that an SARI patient would die with COVID-19. Due to substantial amount of missing vaccination data (54% of all SARI patients with symptoms onset before 6 September, reaching as high as 87% before 1 March), and uncertainty about the reason the data was missing, we used age group and sex-specific vaccination coverage in the individual's municipality of residence as a proxy. For each SARI patient, we used their age group, the socio-economic conditions and vaccination coverage of their municipality of residence. We also included a binary indicator for whether the SARI cases' symptoms began before a substantial amount of vaccination was administered to their age group (ie, if the symptoms occurred before half of the relevant group had been vaccinated). Online supplemental information 1 has details on the model. We

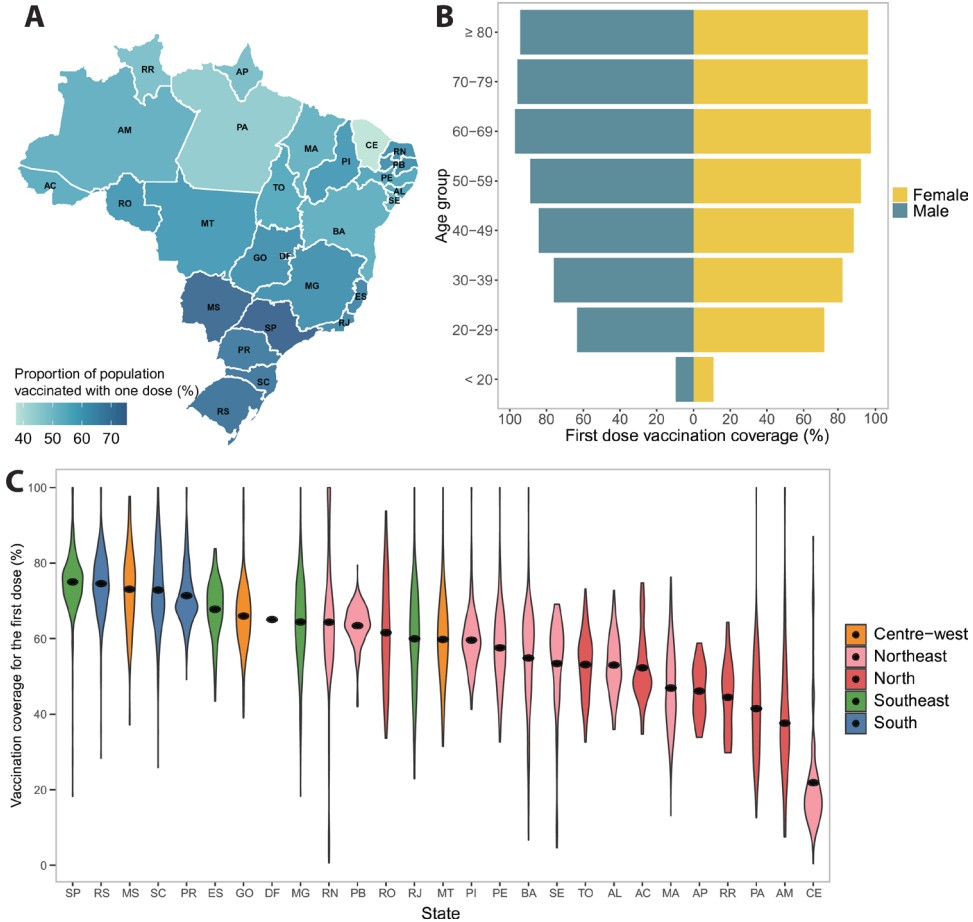

**Figure 1** Inequalities in COVID-19 vaccination coverage across Brazil from 17 January 2021 to 6 September 2021. (A) Percentage of population vaccinated with at least one dose across states as of 6 September 2021. (B) Distribution of vaccination uptake for the first dose between men and women by age group. (C) Vaccination coverage across municipalities (black dot represents the mean coverage).

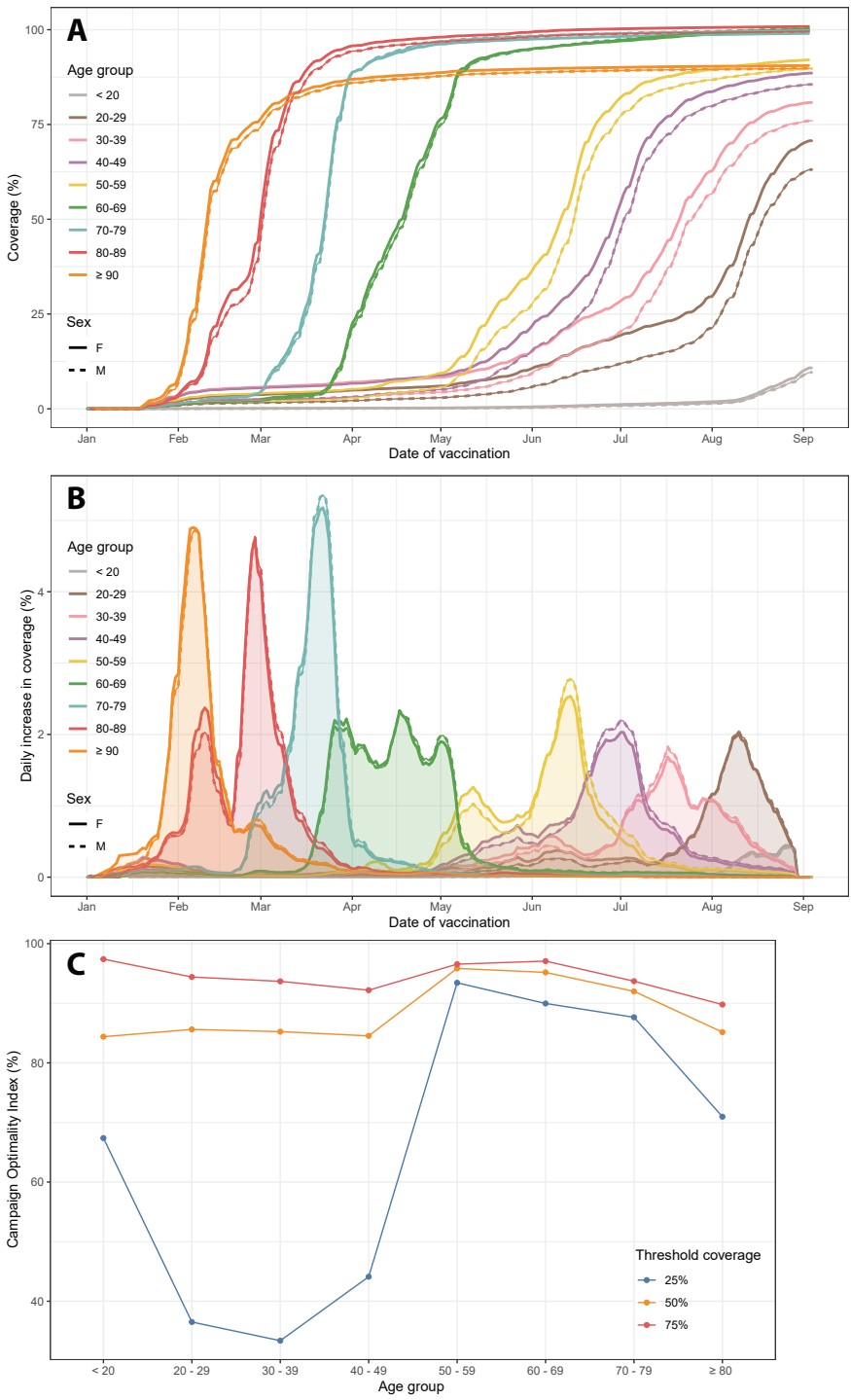

**Figure 2** COVID-19 vaccination uptake over time among age groups. (A) Vaccination coverage for the first dose over time. (B) Daily change in vaccination coverage by age group and sex. (C) Campaign Optimality Index (COI) by age group using threshold vaccination coverage of 25%, 50% and 75%.

computed OR for each regression coefficient $\beta$ as $OR = e^{\beta}$. OR for a continuous covariate can be interpreted as the increased risk of death for a unit increase in the covariate.

### Risk of COVID-19 hospitalisation by age group

We used negative binomial regression (generalised linear model with a log-link function) to model the risk of SARI hospitalisations per person per day in each state adjusting

for age group, whether symptom onset occurred before or after a substantial level of vaccine had been administered (ie, 50% for the age group and state), and each state's average socioeconomic factors such as the level of unemployment, education and per capita income. Given data availability, we used the aggregated values for socioeconomic status across each state as a proxy for the

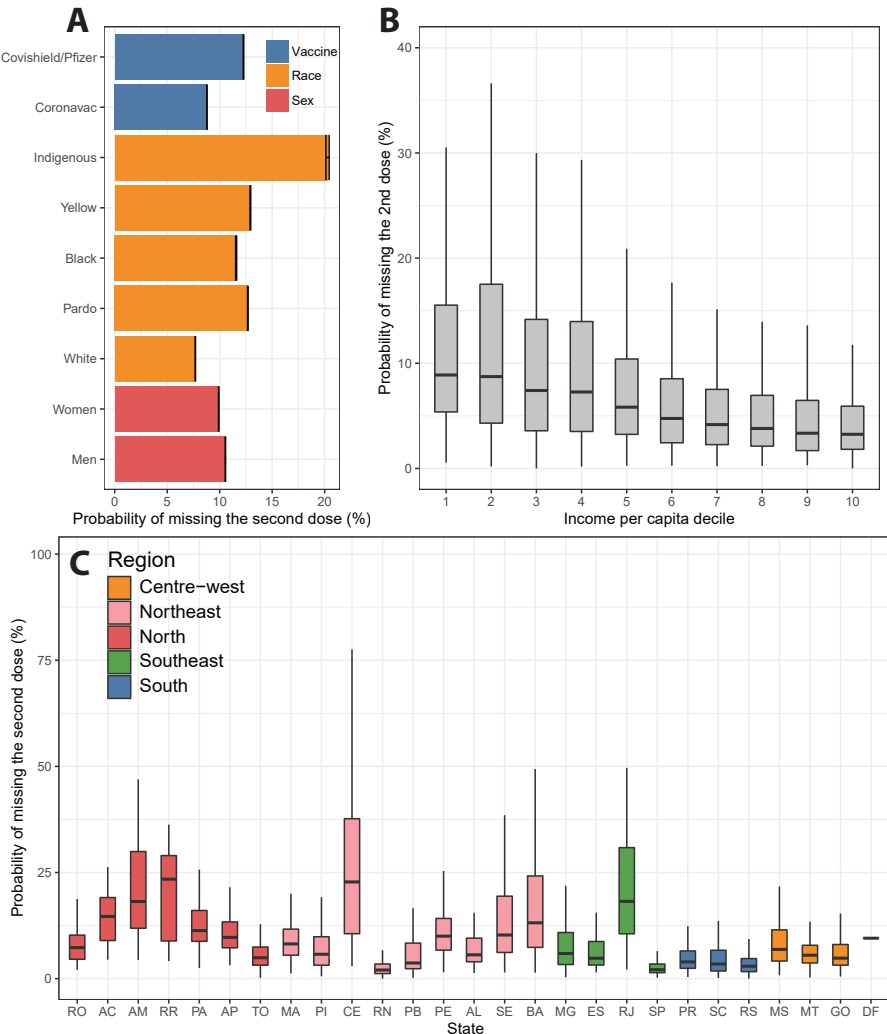

**Figure 3** Probability of missing the second dose of the COVID-19 vaccine. (A) Probability of missing the second dose based on three factors: vaccine type, race and sex. (B) Probability of having a second dose missed calculated at municipality level in terms of the income per capita decile. (C) Probability of missing the second dose calculated for each municipality and grouped by state.

individual's values. Online supplemental information 2 has full details of the model.

### Age distribution of death by vaccination coverage

To assess the potential protective effect of vaccination coverage by age, we looked at the distribution of ages among COVID-19 deaths in the SARI database. We modelled the ages as a sample from a multinomial distribution with parameter $\theta = (\theta_1, \ldots, \theta_n)$ where $\theta_j$ is the probability of a death belonging to the $j$th age group. The posterior distribution of $\theta$ is a Dirichlet distribution with parameters $\alpha = (1 + D_1, \ldots, 1 + D_n)$, where $D_j$ is the number of deaths observed in age group $j$, assuming a non-informative Dirichlet prior distribution with parameters $(1, \ldots, 1)$. Sampling from the posterior distribution provided the marginal credible intervals for the proportion of COVID-19 deaths in each age group at four points in the epidemic:

before 17 January, 17 January to 16 March, 17 March to 16 May and 17 May 2021 to 16 September 2021.

### RESULTS
#### Geographical inequalities in vaccination coverage

Between 17 January 2021 and 6 September 2021, 133.4 million people were recorded as having had a first dose, corresponding to 84.4% of the population above 18 years of age. We determined the proportion of the population vaccinated with at least one dose in each municipality and mapped the distribution of coverage across states (figure 1A). Similar trends can be observed for the proportion of the population that is fully vaccinated (online supplemental figure S1). At the national level, older age groups have higher vaccination coverage, which decreases with age (figure 1B). A greater proportion of women were vaccinated than men; this difference is more evident among younger age groups below the age

of 50 years. Figure 1C shows that there is a clear regional divide in vaccination coverage, with the South and Southeast states having the highest coverage, and the North and Northeast states having the lowest coverage, although there is a substantial amount of variability between the municipalities within each state.

## Premature advancement of the vaccination campaign towards younger groups

Figure 2A shows the proportion of the population vaccinated with a first dose in Brazil, by age group and sex over time. Due to uncertainty in total population size, coverage exceeds 100% for some groups. Similarly, coverages smaller than 100% may not imply the remaining population is not immunised. At the start of the campaign, both healthcare workers and older populations were prioritised for vaccination, hence delayed vaccination for younger groups. The vaccination coverage by sex shows the coverage for men remained consistently lower than women in all age groups over time.

We observed an unanticipated increase in coverage among younger age groups before older groups were fully vaccinated (figure 2B). This pattern is accentuated in populations below 60 years of age. Using the COI, we defined the coverage of older groups when the coverage of younger groups reached a threshold coverage of 25%, 50% and 75%. Larger values indicate a more staggered distribution indicating stronger prioritisation towards older age groups, while lower values indicate that age groups are being vaccinated together (ie, weaker prioritisation). In figure 2C, COI is higher for age groups above 50 years old and drops substantially for younger groups. The difference between age groups is greatest for the threshold of 25% and decreases with higher thresholds, with COI being above 80% for all age groups for the threshold of 75%. Even though the vaccination campaign for younger groups started prematurely, the coverage of younger groups crossed the threshold of 75% when coverage for older groups was higher than 80%.

## Probability of missing the second dose varies by sex, race and income

By 6 September 2021, 11.9% and 10.2% of the individuals vaccinated with the first dose had their second dose delayed and missed, respectively. Only 14.3% of individuals that did not take the second dose after 30 days past the recommended date ended up taking a delayed second dose. Among the individuals that were vaccinated with a first dose of Coronavac, 8.78% (95% CI 8.77% to 8.80%) missed their second dose. This is low compared with the 12.27% of individuals that missed the second dose (95% CI 12.25% to 12.29%) of other vaccines (Covishield and Pfizer), where the optimal interval between the first and second dose is 90 days.

The probability of missing the second dose by race and sex is shown in figure 3A. Men were more likely to miss (relative risk (RR) 1.065, 95% CI 1.063 to 1.067) the second dose than women. Indigenous Brazilians were

also 2.64 (95% CI 2.62 to 2.66) times more likely to miss the second dose than White Brazilians, followed by East Asian Brazilians (RR 1.68, 95% CI 1.68 to 1.69), Pardos (RR 1.65, 95% CI 1.65 to 1.65) and Black Brazilians (RR 1.51, 95% CI 1.50 to 1.51).

The probability of missing the second dose varies with income per capita (figure 3B). Individuals in municipalities with a lower income per capita are more likely to miss their second dose. The median income per capita for municipalities with a 2% probability of skipping the second dose is US$394.95 purchasing power parity (PPP) 2010 (IQR 303.75–468.66). This decreases to US$194.81 PPP 2010 (IQR 153.17–303.17) when the probability of skipping is more than 10%. Figure 3C shows the variation in the probability of missing the second dose by state.

## Vaccination decreases age-adjusted risk of hospitalisation and death

Vaccination decreases the risk of hospitalisation or death due to COVID-19 after adjusting for age group and socioeconomic conditions. Using the age group, 20 years of age and under as a baseline, the risk of COVID-19 death increased with age. For instance, the odds of death are almost 10 times greater for someone above the age of 50 years (OR 9.7 to 29.0, 95% C 9.4 to 29.9) than an individual under 20 years of age. In general, patients living in municipalities with higher first-dose vaccination coverage had lower risk of death (OR 0.820, 95% CI 0.812 to 0.827). Patients living in municipalities with higher proportion of primary education attainment (OR 0.231, 95% CI 0.217 to 0.246) had reduced risk of death. Patients from municipalities with higher unemployment rates were at a higher risk of death (OR 2.29, 95% CI 2.14 to 2.45). An increase of income per capita of US$100 PPP decreased the risk of death (OR 0.961, 95% CI 0.960 to 0.963). Our sensitivity analysis at the individual level, which includes only patients with a recorded vaccination status, showed that patients were at a lower risk of death after being vaccinated with at least one dose. While similar patterns for age groups were observed for hospitalisations, a higher proportion of primary education attainment was associated with higher risk of hospitalisations. Detailed information on model outputs can be found in online supplemental information 3, tables S1 and S2 .

The number of hospitalisations and deaths started decreasing after vaccination was introduced (figure 4A). To verify the relationship between vaccination coverage and risk of death by age groups, we plotted the probability of a patient dying from COVID-19 being in a particular age group prior to 17 January 2021 and after the start of the vaccination campaign (including and after 17 January 2021) (figure 4B). There is a higher probability that those dying before the start of the vaccination campaign were from an older age group. After vaccination began, those dying were more likely to be from age groups with ages below 60 years. This can be attributed to vaccination, where the proportion of vaccination is much higher among people in older age groups due to earlier

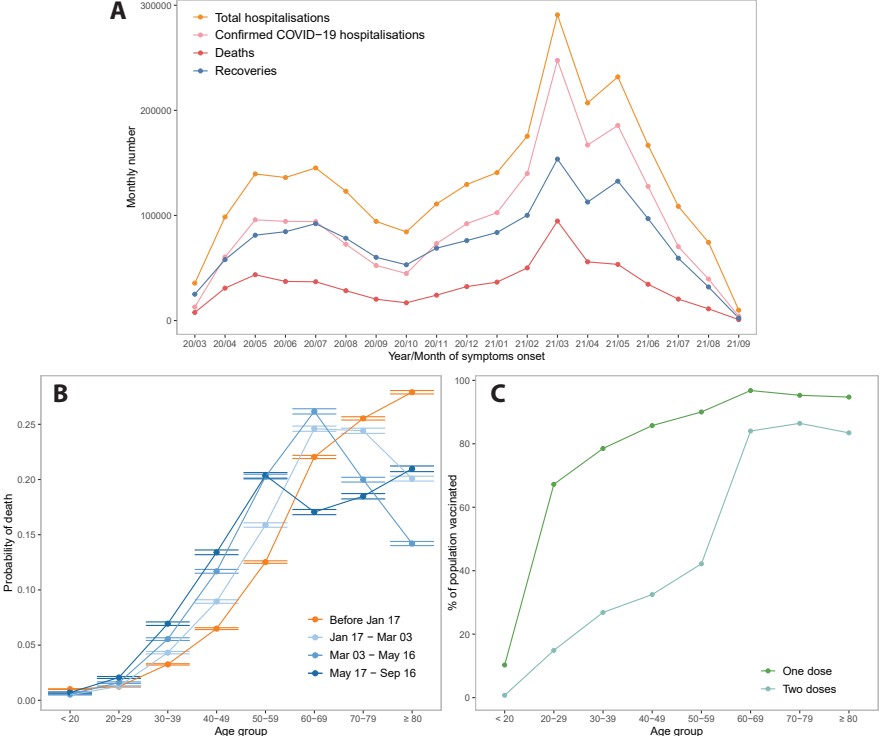

**Figure 4** Risk of SARI death and vaccination coverage by age distribution. (A) Distribution of SARI hospitalisations and deaths over time (March 2020–September 2021). (B) The distribution of age among SARI-linked deaths at four points in time indicates that as the epidemic progressed SARI deaths were more likely to be younger individuals. (C) Proportion of population vaccinated with at least one dose by age group during study period. SARI, severe acute respiratory illness.

uptake (figure 4C). However, individuals under 20 years of age showed a reverse trend, where the proportion of deaths was higher prior to the vaccination campaign.

## DISCUSSION

The Brazilian COVID-19 vaccination campaign led to a decrease in severe cases and deaths. This is further supported by real-world cohort data (Santos et al,[15]). Notwithstanding, the campaign faced issues with rollout delays and vaccine shortages which effected timing and implementation,[16] curtailing effectiveness. Inequalities exist in COVID-19 vaccination coverage, especially between states and different age, sex and socioeconomic groups. Among patients diagnosed with COVID-19, the risks of hospitalisation and death are significantly greater among older individuals and those with lower socioeconomic status. Increasing vaccination reduces the risk of death, especially among older age groups. Vaccine uptake is greater in areas where hospitalisation rates were greater, suggesting better access to healthcare and improved reporting of infections.

Variation in vaccination between population groups was mainly driven by the unequal distribution of vaccines between states. States from the North and Northeast regions, such as Amazonas, Amapá and Ceará, were prioritised for vaccination in January 2021 due to high transmission rates of the P.1 (Gamma) variant, which was first detected in Manaus (the capital

of Amazonas) in January 2021 (Buss et al[17], 2021). Despite this, lower overall coverage for the first dose was observed in the North and Northeast regions of Brazil. These regions are economically less developed and have experienced scare hospital resources during the pandemic (Rocha et al[18], 2021). Delays in notification of vaccinations had an important impact on coverage between states. Even though we determined a 37.5% first-dose coverage for the northeastern state of Ceará, official statistics obtained directly from the Brazilian state secretariats on 6 September 2021 found this was 59.2%. The low COI obtained with a threshold of 25% for age groups younger than 50 years shows the vaccination campaign started prematurely for these groups, which may have impeded vaccination of older groups. It is worth noting that the proposed COI only considers age as a priority criterion and does not include other criteria such as comorbidities and employment (eg, being a healthcare worker).

Inequalities in uptake are evident between the sexes. Women were more likely to have received both doses than men. This may be owing to the early stages of the vaccination campaign, when healthcare workers were prioritised: women make up a significant proportion of healthcare workers in Brazil, accounting for 62.5% of all vaccinated healthcare workers, which corresponds to 10.6% of all vaccinated individuals in our dataset. Moreover, women seek out healthcare

more frequently than men. Based on a survey study with over 170 000 responses, men showed greater hesitancy towards COVID-19 vaccines, indicating the main reason being fear of severe clinical reactions.[19] We also found a longer delay for taking the second dose in municipalities with lower income per capita, which aligns with previous findings that low incomes were associated with higher rates of vaccine hesitancy during the pandemic.[19] This has implications for death risk, as our findings show that SARI patients from poor municipalities are more likely to die from COVID-19 than those from wealthier municipalities.

Vaccination reduces the risk of hospitalisation and death. The positive association between education and hospitalisation could suggest greater accessibility to healthcare in areas with more COVID-19 infection awareness. To further investigate whether vaccination can prevent COVID-19-related deaths for each age group, we considered the age distribution of those that died, across multiple periods of time. We observed a reduction in the proportion of mortality in older age groups once vaccination became available. This trend was the least evident with young people under the age of 20. This was likely owing to the delayed vaccination of young people until July 2021. Due to a lack of complete population-level data, it may be premature to surmise the true protective effects of vaccination coverage on younger age groups. While the overall risk of death is lower for young people, COVID-19 could still pose a threat, regardless of pre-existing medical conditions.[20 21]

Our estimates of vaccination coverage by state and age group are limited by the accuracy of the projected population estimates. Our data for socioeconomic conditions came from the 2011 census, since the 2020 census was not collected and the delayed 2022 census results are not yet available. Further, current socioeconomic conditions may have diverged within states since 2011 due to the variation in state-level funding for social welfare in Brazil.[22] Comprehensive vaccination data of individual SARI patients was not available so we had to rely on overall vaccination coverage for each age group and municipality; this may have introduced discrepancies in the modelled relationship between vaccination coverage and the risk of death. Due to these limitations, we were limited to an ecological study. Human error affects the accuracy of vaccination records (eg, date of vaccination could be replaced by the date when the information was entered into the system rather than administration date). Moreover, the lack of healthcare resources in some areas (eg, Ceará) may have led to lower levels of case notification and notification delay. Self-identification of race may be prone to bias, with educated Black Brazilians more inclined to self-identify and contribute race information than Black Brazilians with low educational attainment.[23]

Although there exists some vaccine hesitancy among certain demographic groups, the overall intention to be vaccinated against COVID-19 is high.[19 24] Vaccine hesitancy is a complex issue; a survey from late 2020 found that when certain countries of origin for the vaccine were mentioned (eg, China and Brazil), the willingness to vaccinate among Brazilians decreased. However, our study shows that individuals that were vaccinated with the first dose of Coronavac were less likely to skip the second dose when compared with individuals that were vaccinated with other vaccines, such Covishield or Pfizer. This may be partially attributed to the fact that the optimal interval between the first and second doses for most vaccines is 90 days, while the interval for Coronavac is only 21–28 days.[25] The disparities in vaccination coverage may have amplified the inequalities in risk of death during the epidemic in Brazil, as the Gamma variant is associated with an increased infection fatality rate.[8]

Despite the allocation of vaccines nationwide and the consideration of priority groups, it is evident from our study that there were several inequality-generating processes within the Brazilian COVID-19 vaccination programme. In general, national vaccine programmes typically focus on controlling for medical conditions between population groups.[26] Our findings demonstrate that to reduce health inequalities, equitable vaccine distribution should address demographic and socioeconomic inequalities.[27] To tackle inequality in uptake, vaccination campaigns should aim to understand the mechanisms that lead to vaccine hesitancy. For instance, reasons for missing the second dose could be attributed to perceived politicisation of the campaign, distrust in pharmaceutical services, and the lack of awareness of the long-term efficacy in Brazil.[15]

The third booster dose has been controversial, as parts of the population were still awaiting their second dose. Vaccination has a protective effect on populations but the details of how this is communicated to the public are complex. Our findings on inequalities in coverage and on demographic groups that are more willing to miss the second dose should be considered when designing media campaigns and interventions to increase vaccine access and uptake.

**Author affiliations**
[1]School of Geography, University of Nottingham, Nottingham, UK
[2]School of Geography and the Environment, University of Oxford, Oxford, UK
[3]Department of Electronic Systems Engineering, University of São Paulo, São Paulo, Brazil
[4]School of Mathematics and Statistics, The University of Melbourne, Melbourne, Victoria, Australia
[5]Department of Zoology, University of Oxford, Oxford, UK
[6]Oxford School of Global and Area Studies and Latin American Centre, University of Oxford, Oxford, UK
[7]King's Brazil Institute, King's College London, London, UK
[8]Departamento de Molestias Infecciosas e Parasitarias & Instituto de Medicina Tropical da Faculdade de Medicina, University of São Paulo, São Paulo, Brazil
[9]School of Mathematical Sciences, University of Southhampton, Southhampton, UK
[10]Oxford School of Global and Area Studies, University of Oxford, Oxford, UK

**Contributors** SLL is the guarantor and accepts full responsibility for the finished work and the conduct of the study, had access to the data, and controlled the decision to publish. SLL: conceptualisation, data curation, formal analysis,

investigation, methodology, software, validation, visualisation, writing–original draft preparation, writing–review and editing. CAPJ: conceptualisation, data curation, formal analysis, investigation, methodology, software, validation, visualisation, writing–original draft preparation, writing–review and editing. AZ: conceptualisation, formal analysis, methodology, software, validation, writing–original draft preparation, writing–review. AS: investigation, validation, writing–review. ES: investigation, validation, writing–review. VN: formal analysis, validation, writing–review. C-HW: formal analysis, validation, writing–review and editing. JM: conceptualisation, writing–review and editing, supervision.

**Funding** SL was supported by the Oxford Martin Programme on Pandemic Genomics and the Canadian Social Sciences and Humanities Research Council (SSHRC) Doctoral Fellowship. CAPJ is supported by FAPESP (2019/21858-0 and 2022/15985-1). AZ was supported by the Oxford Martin School Programme on Pandemic Genomics. CAPJ and VN were supported by Coordenação de Aperfeicoamento de Pessoal de Nível Superior–Brasil (CAPES)–Finance Code 001. VN is supported by the Brazilian National Council for Scientific and Technological Development (CNPq: 304714/2018-6). ES is supported by a UKRI Medical Research Council-São Paulo Research Foundation (FAPESP) CADDE partnership award (MR/S0195/1 and FAPESP 18/14389-0) (http://caddecentre.org/). JPM is supported by the Oxford Martin School Programme on Pandemic Genomics.

**Competing interests** None declared.

**Patient and public involvement** Patients and/or the public were not involved in the design, or conduct, or reporting, or dissemination plans of this research.

**Patient consent for publication** Not applicable.

**Provenance and peer review** Not commissioned; externally peer reviewed.

**Data availability statement** Data are available on reasonable request.

**ORCID iDs**
Sabrina L Li http://orcid.org/0000-0002-1183-126X
Carlos A Prete Jr http://orcid.org/0000-0002-3907-423X
Alexander E Zarebski http://orcid.org/0000-0003-1824-7653
Andreza Aruska de Souza Santos http://orcid.org/0000-0003-3585-8683
Ester C Sabino http://orcid.org/0000-0003-2623-5126
Vitor H Nascimento http://orcid.org/0000-0002-0543-4735
Chieh-Hsi Wu http://orcid.org/0000-0001-9386-725X
Jane P Messina http://orcid.org/0000-0001-7829-1272

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
