## [Reviewer comments · BMJ Open]

ARTICLE DETAILS

TITLE (PROVISIONAL)	The Brazilian COVID-19 vaccination campaign: A modelling analysis of socio-demographic factors on uptake
AUTHORS	Li, Sabrina; Prete Junior, Carlos Augusto; Zarebski, Alexander; Santos, Andreza; Sabino, Ester; Nascimento, Vitor; Wu, Chieh-Hsi; Messina, Jane

VERSION 1 – REVIEW

REVIEWER	Bolcato, Matteo University of Padua Via Falloppio
REVIEW RETURNED	19-Jun-2023

GENERAL COMMENTS	I have carefully read the complex study presented in the article, I am convinced that it is very important and useful on a global level. The same shows with consistent methodology that the vaccination inequality given by different economic and social factors causes a proportional damage from the point of view of public health. So I think it deserves publication. If possible, I would like to suggest some ideas: - is it possible to hypothesize some practical element to eliminate or decrease inequality in your context? - Can interfering factors be excluded from the efficacy results?- the authors could briefly underline the ethical importance of vaccination equity see epr example cit doi: 10.3390/vaccines9060538.
---

REVIEWER	Bastos, Leonardo Fundacao Oswaldo Cruz, Scientific Computing Program
REVIEW RETURNED	22-Aug-2023

GENERAL COMMENTS	The manuscript "The Brazilian COVID-19 vaccination campaign: A modelling analysis of socio-demographic factors on uptake" present a set of ecological analysis to show the impact of the vaccination inequality in Brazil confronting vaccination data individual level data with hospitalization individual data, socioeconomic and vaccination coverage at municipality level. The study is an ecological study because socioeconomic can not be attributed to an individual also the hospitalization status in the vaccination database is unknown, so the vaccination coverage was used as a proxy. It is very important to highlight the impact of vaccination inequality, since it does cost lives. I have several comments for authors. Even though there are indeed limitations on the vaccination leading to inequality, it is important to emphasises that there is evidence that the vaccination in Brazil were effective. For instance, there are
---

some papers showing the impact of the COVID-19 vaccination against severe cases and death in Brazil. Ferreira et al (2023) presented a counterfactual analysis to evaluate the impact of implementation and timing of the COVID-19 vaccination, and Dos Santos et al. (2023) presented a large vaccine effectiveness study in Brazil showing real-world evidence of effectiveness of COVID-19 vaccination in Brazil.

The authors choose to consider just the first two doses, and we know that a booster was needed to confer more protection against a severe COVID case. On the other hand, the inequality for boosters would be greater. I wonder if the authors could include at least one booster in the analysis. The data is available, the time frame could be extended. Why the choice of just a couple of months, as I said the datasets used are freely available and have more than two years of information since the first vaccine (January 2021).

I would like to thank the editor for the opportunity and I have some other major and minor comments here:

Lines 65 and 66 (Introduction). "Given the increasing risk of SARS-CoV-2 infection and death with age (O'Driscoll et al. 2021)...". I am not convinced there is a different infection risk according to age. It is an airborne virus so in theory everyone taking the same measures of protection, masks for instance, would be equally likely to be infected. On the other hand, there is a clear increasing risk of hospitalization and death due to COVID-19 for older groups. In fact there is a manuscript with the same claim using Brazilian data (Lana et al. 2021) and their analyses were taken into account by the Brazilian Ministry of Health as part of the decision making process for the vaccination campaign.

Lines 93-96 (Methods, Data sources). I am not sure what SRAG database is. I am aware that SRAG is the Portuguese version of SARI. The SARI data is actually from the Influenza epidemiological surveillance information system (SIVEP-Gripe) which records hospitalizations and deaths with SARI symptoms from the whole country.

Lines 99-102 (Methods, Data sources). I think it is a good idea to include confirmed COVID-19 cases and also SARI cases without lab confirmation as possible COVID-19 cases. However the justification is not optimal. Lack of tests for hospitalized cases was not a big issue, because since start of the epidemic the priority for testing was given to hospitalized cases as part of an universal Influenza surveillance system. So, in theory, all hospitalized cases were tested. The socioeconomic bias is valid for mild cases which indeed suffer from lack of tests. For hospitalized cases the main problem that could justify the use of non-confirmed SARI cases as possible COVID-19 cases was the moment the patient was tested. Several patients were admitted in hospital out of the positivity window, therefore the most likely result was a negative result.

Lines 107-113 (Socioeconomic conditions and population size) The data used may be too old. I am aware that it is the most up-to-date information available since there was no census in 2020, and the results of the delayed 2022 census are not yet available. So it would be important to add a limitation in the discussion mentioning that the socioeconomic estimates and the population projections

used may diverge from what actually happened in 2020-2021. In particular, the small area projections may lead to different estimates of the non-vaccinated groups. For instance, a large population projection error could lead to a situation where the number of vaccinated people for a particular municipality and age group would be greater than the project population for that municipality and age-group. In fact, the preliminary results of 2022 census has shown that population projections for some municipalities were overestimated. In the Results section, lines 200-202, the error on population projections was briefly mentioned. I think this problem should be mentioned in the methods, pointed out in the results and the implications discussed in the last section.

Lines 161-162. (Risk of COVID-19 hospitalisation by age group)
"We used quasi-Poisson regression (GLM with a log-link function) to model the number of SARI hospitalisations...". A quasi-Poisson regression is not a generalised linear model (GLM), it uses the GLM framework to do the inference. A GLM assumes a probability distribution for the outcome, i.e. a likelihood function. A quasi-Poisson regression is a generalization of the Poisson model but instead of using a Poisson likelihood it uses a quasi-likelihood function. See McCullagh and Nelder (1989, Chapters 6 for Poisson and 9 for quasi-likelihood functions).

Figure 2C. In the Figure title there is a comment "This index could not be calculated for the age group 20–29 using the threshold 75% because this group had not reached a coverage of 75% by September 6th 2021." which seems fair, but the COI value for the age group 20-29 using the threshold 75% is in the plot.

Lines 242-247 (Results). Are those values correct? The minimum wage in Brazil was around R\$1200.00. The values presented as median income per capita by municipalities seems quite low. Why is that occurring? In the Figure 3B the income per capita were partitioned by deciles. I am not sure if I understood. In Figure 3B, each the boxplot presents all estimated probabilities for a given decile, whereas in the text given the author present the range of values income per capita that leads to a fixed probability of skipping second dose. Anyway, this part should become clearer.

Minor:

Methods. I found a lack of consistency in the choice of inference approaches. The methods used are correct, but each method follow a different inference path. In subsection "Risk of COVID-19 death by age-group" an usual frequentist logist model was used. Then in the next subsetion "Risk of COVID-19 hospitalisation by age group" a quasi-Poisson was used, which still falls into a frequentist model but, as mentioned before, doesn't have a likelihood. And last, in the subsection "Age distribution of death by vaccination coverage" the authors use a multinomial-dirichlet Bayesian model. Again, the methods are correct, I was just expecting the same inference approach.

Typos:

	Line 40: "COVID-19". Line 99: "dyspnea". Line 290: "risk of" References: Ferreira et al (2023) https://pubmed.ncbi.nlm.nih.gov/36439909/ Lana et al. (2021) https://pubmed.ncbi.nlm.nih.gov/34644749/ Dos Santos et al. (2023) https://pubmed.ncbi.nlm.nih.gov/36936517/
--	---

VERSION 1 – AUTHOR RESPONSE

Reviewer: 1 Dr. Matteo Bolcato, University of Padua Via Falloppio

- 1. I have carefully read the complex study presented in the article, I am convinced that it is very important and useful on a global level. The same shows with consistent methodology that the vaccination inequality given by different economic and social factors causes a proportional damage from the point of view of public health. So I think it deserves publication.**

Thank you very much for your review and comments about the paper.

- 2. If possible, I would like to suggest some ideas: - is it possible to hypothesize some practical element to eliminate or decrease inequality in your context?**

A positive call to action is a good suggestion. In the Discussion, we have now elaborated on some additional discussion points on the topic of inequality of uptake stemming from Brazil's COVID-19 Vaccination Campaign:

Page 18, lines 360-380, "Despite the allocation of vaccines nationwide and the consideration of priority groups, it is evident from our study that there were several inequality-generating processes within the Brazilian COVID-19 vaccination programme. In general, national vaccine programmes typically focus on controlling for medical conditions between population groups (Mamelund and Dimka, 2021). Our findings demonstrate that to reduce health inequalities, equitable vaccine distribution should address demographic and socioeconomic inequalities (Rydland et al., 2022). To tackle inequality in uptake, vaccination campaigns should aim to understand the mechanisms that lead to vaccine hesitancy. For instance, reasons for missing the second dose could be attributed to perceived politicization of the campaign, distrust in pharmaceutical services, and the lack of awareness of the long-term efficacy in Brazil (K.C.O. Dos Santos et al., 2023)."

References:

- Mamelund, S.-E., Dimka, J., 2021. Social inequalities in infectious diseases. *Scand. J. Public Health* 49, 675–680. <https://doi.org/10.1177/1403494821997228>
- Rydland, H.T., Friedman, J., Stringhini, S., Link, B.G., Eikemo, T.A., 2022. The radically unequal distribution of Covid-19 vaccinations: a predictable yet avoidable symptom of the fundamental causes of inequality. *Humanit. Soc. Sci. Commun.* 9, 1–6. <https://doi.org/10.1057/s41599-022-01073-z>
- Dos Santos, K.C.O., Junqueira-Marinho, M. de F., Reis, A.T., Camacho, K.G., Nehab, M.F., Abramov, D.M., Azevedo, Z.M.A. de, Menezes, L.A. de, Salú, M. dos S., Figueiredo, C.E. da S., Moreira, M.E.L., Vasconcelos, Z.F.M. de, Carvalho, F.A.A. de, Mello, L. de R. de, Correia, R.F., Gomes Junior, S.C. dos S., Moore, D.C.B.C., 2023. Social Representations of Hesitant Brazilians about Vaccination

against COVID-19. *Int. J. Environ. Res. Public. Health* 20, 6204.

<https://doi.org/10.3390/ijerph20136204>

3. Can interfering factors be excluded from the efficacy results?

Thank you for your comment. Our model results include several socioeconomic variables that are used to account for potential confounding effects on the risk of hospitalisation and death. We include these variables as each acts as a metric of the Social Determinant of Health (Marmot et al., 2005), which is used to characterize potential impacts of inequality. These variables were extracted from the Brazilian census at the municipality level. We do not draw a causal inference from our statistical models as data are not available at the individual level. In our hospitalisation model, we have now excluded unemployment due to its strong correlation with income per capita at the state level as evidenced by a large VIF, see response to Reviewer 2 regarding statistical models.

References

Marmot, M., & Wilkinson, R. (Eds.). (2005). *Social determinants of health*. OUP Oxford.

4. The authors could briefly underline the ethical importance of vaccination equity see epr example cit doi: 10.3390/vaccines9060538.

<https://www.mdpi.com/2076-393X/9/6/538> <-- vaccines paper

We have now addressed this in our response to Reviewer 2, comment 5.

Reviewer: 2 Prof. Leonardo Bastos, Fundacao Oswaldo Cruz

5. The manuscript "The Brazilian COVID-19 vaccination campaign: A modelling analysis of socio-demographic factors on uptake" present a set of ecological analysis to show the impact of the vaccination inequality in Brazil confronting vaccination data individual level data with hospitalization individual data, socioeconomic and vaccination coverage at municipality level. The study is an ecological study because socioeconomic can not be attributed to an individual also the hospitalization status in the vaccination database is unknown, so the vaccination coverage was used as a proxy. It is very important to highlight the impact of vaccination inequality, since it does cost lives. I have several comments for authors.

Thank you for reviewing our study and providing many constructive suggestions. We have now emphasized the importance of vaccine equity in the Introduction:

Page 3, lines 56-57: "Equitable and ethical administration of vaccines should be strived for because of its impact on reducing mortality in vulnerable populations, however this remains an outstanding challenge (Bolcato et al., 2021; Ye et al., 2022)."

We have also stressed that data limitations only enabled us to conduct an ecological study in the Discussion:

Page 17, lines 350-353: "Due to these limitations, we were limited to an ecological study. Human error affects the accuracy of vaccination records (e.g. date of vaccination could be replaced by the date when the information was entered into the system rather than administration date)"

In response to Comment 2 from Reviewer 1, we added some discussion on how vaccination inequality can be tackled.

References

Bolcato, M., Rodriguez, D., Feola, A., Di Mizio, G., Bonsignore, A., Ciliberti, R., Tettamanti, C., Trabucco Aurilio, M., Aprile, A., 2021. COVID-19 Pandemic and Equal Access to Vaccines. *Vaccines* 9, 538. <https://doi.org/10.3390/vaccines9060538>

Ye, Y., Zhang, Q., Wei, X., Cao, Z., Yuan, H.-Y., Zeng, D.D., 2022. Equitable access to COVID-19 vaccines makes a life-saving difference to all countries. *Nat. Hum. Behav.* 6, 207–216.

<https://doi.org/10.1038/s41562-022-01289-8>

6. Even though there are indeed limitations on the vaccination leading to inequality, it is important to emphasises that there is evidence that the vaccination in Brazil were effective. For instance, there are some papers showing the impact of the COVID-19 vaccination against severe cases and death in Brazil. Ferreira et al (2023) presented a counter factual analysis to evaluate the impact of implementation and timing of the COVID-

19 vaccination, and Dos Santos et al. (2023) presented a large vaccine effectiveness study in Brazil showing real-world evidence of effectiveness of COVID-19 vaccination in Brazil.

We have now partially addressed this in the Introduction (See comment 5).

In the Results (**page 15, Figure 4**), our analysis highlights the effectiveness of the vaccination campaign on preventing deaths, especially among elderly age groups. This is further discussed at the start of the Discussion, where we have now added the following:

Page 15, lines 291-295: "The Brazilian COVID-19 vaccination campaign led to a decrease in severe cases and deaths. This is further supported by real-world cohort data (C.V.B Dos Santos et al., 2023). Notwithstanding, the campaign faced issues with rollout delays and vaccine shortages which effected timing and implementation (Ferreira et al., 2023), curtailing effectiveness".

7. The authors choose to consider just the first two doses, and we know that a booster was needed to confer more protection against a severe COVID case. On the other hand, the inequality for boosters would be greater. I wonder if the authors could include at least one booster in the analysis. The data is available, the time frame could be extended. Why the choice of just a couple of months, as I said the datasets used are freely available and have more than two years of information since the first vaccine (January 2021).

Thank you for this suggestion. We did not consider the booster dose because it is out of the scope of our study aims. Our study focuses on the uptake of doses by September 2021, prior to the introduction of the booster. By this date, the majority of the population had received 2 doses, which allows us to make more robust inferences about the prevention effects of the vaccination campaign on COVID-19 within the population.

We excluded the booster for the following reasons: (1) A key focus of our paper is to calculate the proportion of the population that missed the first dose. This is critical for understanding behavioral elements that may influence future uptake of the vaccine.(2) Our paper aims to understand the mechanisms of vaccination uptake, or, the potential factors that influence the likelihood of people getting vaccinated. We are already able to address this with our analysis of doses 1 and 2 where data are more complete. (3) The time frame for taking the booster varied for each age and priority group, and did not follow the same consistent time window prescribed for doses 1 and 2. Therefore, it would be difficult to compare the findings drawn from the booster analysis with our analysis of the initial vaccination campaign.

8. I would like to thank the editor for the opportunity and I have some other major and minors comments here: Lines 65 and 66 (Introduction). "Given the increasing risk of SARS-CoV-2 infection and death with age (O'Driscoll et al. 2021)...". I am not convinced there is a different infection risk according to age. It is an airborne virus so in theory everyone taking the same measures of protection, masks for instance, would be equally likely to be infected. On the other hand, there is a clear increasing risk of hospitalization and death due to COVID-19 for older groups. In fact there is a manuscript with the same claim using Brazilian data (Lana et al. 2021) and their analyses were taken into account by the Brazilian Ministry of Health as part of the decision making process for the vaccination campaign.

We agree that infection risk does not necessarily *increase* with age, and have now revised the text to make this point more precise, we emphasise that infection risk *changes* with age. Previous studies on COVID-19 in Brazil have found that the attack rate of SARS-CoV-2 varies with age (Prete Jr. et al. 2022). Lana et al. (2021) also found that vaccinating elders reduces both the risk of infection and death.

In the Introduction, this section now reads:

Page 3, lines 57-67: "...During the campaign, dose allocation was determined by the total population of each state without adjusting for demographic differences. While older age is associated with varying risk of SARS-CoV-2 infection (Prete et al., 2022), it is associated with increasing risk of hospitalisation and death (Lana et al. 2021; O'Driscoll et al. 2021). Thus, understanding vaccine uptake across age groups is critical for reducing mortality and health inequalities and optimizing the distribution of limited vaccine doses. More importantly, little is known about how the vaccination programme impacts age-specific hospitalisations and COVID-19- related death rates(Hitchings et al.

2021). To optimise vaccine distribution, further investigation is needed to understand the age-specific reduction in the risk of severe COVID-19 and death, especially among those most vulnerable.”

References

Lana, R.M., Freitas, L.P., Codeço, C.T., Pacheco, A.G., Carvalho, L.M.F. de, Villela, D.A.M., Coelho, F.C., Cruz, O.G., Niquini, R.P., Porto, V.B.G., Gava, C., Gomes, M.F. da C., Bastos, L.S., 2021. Identification of priority groups for COVID-19 vaccination in Brazil. *Cad. Saude Publica* 37, e00049821. <https://doi.org/10.1590/0102-311X00049821>

Prete, Carlos A, Jr, Lewis F Buss, Charles Whittaker, Tassila Salomon, Marcio K Oikawa, Rafael H M Pereira, Isabel C G Moura, et al. 2022. “SARS-CoV-2 Antibody Dynamics in Blood Donors and COVID-19 Epidemiology in Eight Brazilian State Capitals: A Serial Cross-Sectional Study.” *eLife* 11. <https://doi.org/10.7554/eLife.78233>

9. Lines 93-96 (Methods, Data sources). I am not sure what SRAG database is. I am aware that SRAG is the Portuguese version of SARI. The SARI data is actually from the Influenza epidemiological surveillance information system (SIVEP-Gripe) which records hospitalizations and deaths with SARI symptoms from the whole country.

We have now revised this by replacing SARI/SRAG with SIVEP-Gripe. We have further defined SARI and clarified this with a link to the data portal: [opendatasus](https://opendatasus.gov.br/). In the Data Sources section under Methods, this now reads:

Page 4, lines 93-103: “We selected SARI hospitalisations and deaths, disregarding non-hospitalised individuals that did not die. All SARI cases confirmed to be caused by other respiratory viruses were excluded from the analysis. We included both confirmed COVID-19 cases and SARI cases with unknown etiology, as those were likely related to COVID-19 but not lab-confirmed (de Souza et al. 2020)(Li et al. 2021). This might be due to two reasons: the low rates of COVID-19 testing in Brazil, especially in poorer areas as shown by de Souza et al. 2020; and high proportion of false negative tests that were likely applied after the time window where SARS-CoV-2 is detectable by PCR. From all SARI cases with unknown or missing etiology with symptom onset during the study period, 24.9% did not have a PCR test result reported in SIPEP-Gripe, and most of the remaining 75.1% had likely a false negative test result.”

10. Lines 99-102 (Methods, Data sources). I think it is a good idea to include confirmed COVID-19 cases and also SARI cases without lab confirmation as possible COVID-19 cases. However the justification is not optimal. Lack of tests for hospitalized cases was not a big issue, because since start of the epidemic the priority for testing was given to hospitalized cases as part of an universal Influenza surveillance system. So, in theory, all hospitalized cases were tested. The socioeconomic bias is valid for mild cases which indeed suffer from lack of tests. For hospitalized cases the main problem that could justify the use of non-confirmed SARI cases as possible COVID-19 cases was the moment the patient was tested. Several patients were admitted in hospital out of the positivity window, therefore the most likely result was a negative result.

Thank you – the comment made about the lack of testing for mild cases is correct. Only 24.9% of all SARI cases reported in 2021 with missing or unknown etiology were not tested by PCR or had a PCR test result that was never reported in the SIVEP-Grip database. Most of the remaining 75.1% patients had likely a false negative test result.

We have now updated the manuscript to clarify the reasons why we consider SARI cases with unknown or missing etiology in our analyses. Please see comment 9 for the changes made in the manuscript.

11. Lines 107-113 (Socioeconomic conditions and population size) The data used may be too old. I am aware that it is the most up-to-date information available since there was no census in 2020, and the results of the delayed 2022 census are not yet available. So it would be important to add a limitation in the discussion mentioning that the socioeconomic estimates and the population projections used may diverge from what actually happened in 2020-2021.

Based on your suggestion, we have now added the following to further highlight this:

Page 17, lines 343-347: "Our data for socioeconomic conditions came from the 2011 census, since the 2020 census was not collected and the delayed 2022 census results are not yet available. Further, current socioeconomic conditions may have diverged within states since 2011 due to the variation in state-level funding for social welfare in Brazil (Stankiewicz et al. 2021)."

References

Stankiewicz Serra, A., Yalonetzky, G. I., & Maia, A. G. (2021). Multidimensional poverty in Brazil in the early 21st Century: Evidence from the Demographic Census. *Social Indicators Research*, 154(1), 79-114. <https://doi.org/10.1007/s11205-020-02568-5>

12. In particular, the small area projections may lead to different estimates of the non-vaccinated groups. For instance, a large population projection error could lead to a situation where the number of vaccinated people for a particular municipality and age group would be greater than the project population for that municipality and age-group. In fact, the preliminary results of 2022 census has shown that population projections for some municipalities were overestimated. In the Results section, lines 200-202, the error on population projections was briefly mentioned. I think this problem should be mentioned in the methods, pointed out in the results and the implications discussed in the last section.

Thank you for this suggestion. We agree that using population projections to characterise vaccination coverage has its limitations. However, in our study, we are mainly interested in using population projections to highlight how vaccination uptake varies based on the distribution of populations across states. Thus, determining absolute coverage is not an essential component of our study. In our analysis of the vaccination campaign index (i.e. the COI), we also use relative vaccination coverage between age groups. This is illustrated in Figure 2A to show how coverage changes over time. It is not used in any analytical analysis.

13. Lines 161-162. (Risk of COVID-19 hospitalisation by age group) "We used quasi-Poisson regression (GLM with a log-link function) to model the number of SARI hospitalisations...". A quasi-Poisson regression is not a generalised linear model (GLM), it uses the GLM framework to do the inference. A GLM assumes a probability distribution for the outcome, i.e. a likelihood function. A quasi-Poisson regression is a generalization of the Poisson model but instead of using a Poisson likelihood it uses a quasi-likelihood function. See McCullagh and Nelder (1989, Chapters 6 for Poisson and 9 for quasi-likelihood functions).

Thank you for this suggestion. While this appears to be an issue of communication rather than content we acknowledge the point. We have now revised our analysis of the hospitalisation data, using a negative binomial GLM to stay within the likelihood framework. To retrieve a rate of hospitalisation per person per day as our outcome, we have also included offsets, for the duration (over which the hospitalisations occurred) and the population of the state. We have revised the Methods and Discussion to reflect this change. Further detail can be found in the Supplementary Information.

14. Figure 2C. In the Figure title there is a comment "This index could not be calculated for the age group 20–29 using the threshold 75% because this group had not reached a coverage of 75% by September 6th 2021." which seems fair, but the COI value for the age group 20-29 using the threshold 75% is in the plot.

We thank the reviewer for raising this. This sentence is wrong and was mistakenly added into the figure caption. We have now removed it from the caption.

Please note that even though it is true that the age group 20-29 did not reach a coverage of 75% by September 6th, we used a relative coverage instead of an absolute coverage to compute the COI. We define the relative coverage at time t as the absolute coverage at time t divided by the absolute coverage on September 6th, 2021. Therefore, all age groups reached a relative coverage of 75% at some point before September 6th, 2021.

15. Lines 242-247 (Results). Are those values correct? The minimum wage in Brazil was around R\$1200.00. The values presented as median income per capita by municipalities seems quite low. Why is that occurring? In the Figure 3B the income per capita were partitioned by deciles. I am not sure if I understood. In Figure 3B, each the boxplot

presents all estimated probabilities for a given decile, whereas in the text given the author present the range of values income per capita that leads to a fixed probability of skipping second dose. Anyway, this part should become clearer.

Thank you – we have revised this section to clarify our usage of income per capita. The income per capita described on Page 11 comes from the 2010 census; we only used this to provide some background context. However in our analysis (shown in Figure 3), we used only income quantiles. To avoid confusion, we have now converted the values from Reais 2010 to US\$ Purchasing Power Parity (PPP) 2010.

Page 12, lines 246-250: “The median income per capita for municipalities with a 2 % probability of skipping the second dose is 394.95 US\$ Purchasing Power Parity (PPP) 2010 (IQR 303.75 - 468.66). This decreases to 194.81 US\$ PPP 2010 (IQR 153.17 - 303.17) when the probability of skipping is more than 10%. Figure C shows the variation in the probability of missing the second dose by state.”

16. Minor: Methods. I found a lack of consistency in the choice of inference approaches. The methods used are correct, but each method follow a different inference path. In subsection "Risk of COVID-19 death by age-group" an usual frequentist logist model was used. Then in the next subsetion "Risk of COVID-19 hospitalisation by age group" a quasi-Poisson was used, which still falls into a frequentist model but, as mentioned before, doesn't have a likelihood. And last, in the subsection "Age distribution of death by vaccination coverage" the authors use a multinomial-dirichlet Bayesian model. Again, the methods are correct, I was just expecting the same inference approach.

Please refer to the comments above regarding replacing the quasi-Poisson regression with a negative binomial GLM to stay with the likelihood framework. Please see our response to comment 13.

We appreciate the disjointedness of moving between frequentist and Bayesian analysis that our use of various inference approaches may be perplexing. However, since these are distinct analyses, the two regressions vs the proportions through time this doesn't seem unreasonable. We've used the Dirichlet credible intervals because it seems to be an easier way to characterise the age distribution of death. Further, with the large amount of data, the credible intervals are small suggesting the results are primarily due to the likelihood (not the prior) so both inference approach would yield similar results, i.e., the confidence intervals would be similar to our credible intervals.

17. Typos:

Line 40: "COVID-19".

Line 99: "dyspnea".

Line 290: "risk of"

Thank you for spotting these typos. These have now been fixed in the manuscript.

References:

Ferreira et al (2023) <https://pubmed.ncbi.nlm.nih.gov/36439909/>

Lana et al. (2021) <https://pubmed.ncbi.nlm.nih.gov/34644749/>

Dos Santos et al. (2023) <https://pubmed.ncbi.nlm.nih.gov/36936517/>